# Submodular Field Grammars: Representation, Inference, and Application to Image Parsing

**Abram L. Friesen** and **Pedro Domingos**
Paul G. Allen School of Computer Science and Engineering
University of Washington
Seattle, WA 98195
`{afriesen,pedrod}@cs.washington.edu`

## Abstract

Natural scenes contain many layers of part-subpart structure, and distributions over them are thus naturally represented by stochastic image grammars, with one production per decomposition of a part. Unfortunately, in contrast to language grammars, where the number of possible split points for a production $A \to BC$ is linear in the length of $A$, in an image there are an exponential number of ways to split a region into subregions. This makes parsing intractable and requires image grammars to be severely restricted in practice, for example by allowing only rectangular regions. In this paper, we address this problem by associating with each production a submodular Markov random field whose labels are the subparts and whose labeling segments the current object into these subparts. We call the resulting model a submodular field grammar (SFG). Finding the MAP split of a region into subregions is now tractable, and by exploiting this we develop an efficient approximate algorithm for MAP parsing of images with SFGs. Empirically, we show promising improvements in accuracy when using SFGs for scene understanding, and demonstrate exponential improvements in inference time compared to traditional methods, while returning comparable minima.

## 1 Introduction

Understanding natural scenes is a challenging problem that requires simultaneously detecting, segmenting, and recognizing each object in a scene despite noise, distractors, and ambiguity. Fortunately, natural scenes possess inherent structure in the form of contextual and part-subpart relationships between objects. Such relationships are well modeled by a grammar, which defines a set of production rules that specify the decomposition of objects into their parts. Natural language is the most common application of such grammars, but the compositional structure of natural scenes makes stochastic image grammars a natural candidate for representing distributions over images (see Zhu and Mumford [1] for a review). Importantly, natural language can be parsed efficiently with respect to a grammar because the number of possible split points for each production $A \to BC$ is linear in the length of the constituent corresponding to $A$. However, images cannot be parsed efficiently in this way because there are an exponential number of ways to split an image into arbitrarily-shaped subregions. As such, previous image-grammar approaches could only ensure tractability by severely limiting the possible decompositions of each region either explicitly, for example by allowing only rectangular regions, or by sampling (e.g., Poon and Domingos [2], Zhao and Zhu [3]).

Due to these limitations, many approaches to scene understanding instead use a Markov random field (MRF) to define a probabilistic model over pixel labels (e.g., Shotton et al. [4], Gould et al. [5]), thereby capturing some natural structure while still permitting objects to have arbitrary shapes. Most such MRFs use planar- or tree-structured graphs in the label space [6, 7]. While these models can improve labeling accuracy, their restricted structures mean that they can capture little of the compositional structure present in natural images without an exponential number of labels. Inference in MRFs is intractable in general [8] but is tractable under certain restrictions. For pairwise binary MRFs, if the energy is submodular [9], meaning that each pair of neighboring pixels prefers to have

the same label – a natural assumption for images – then the exact MAP labeling of the MRF can be efficiently recovered with a graph-cut algorithm [10–12]. For multi-label problems, a constant-factor approximation can be found efficiently using a move-making algorithm, such as $\alpha$-expansion [13].

In this work, we define a powerful new class of tractable models that combines the tractability and region-shape flexibility afforded by submodular MRFs with the high-level compositional structure of an image grammar. We associate with each production $A \rightarrow BC$ a submodular MRF whose labels are the subconstituents (i.e., $B, C$) of that production. We call the resulting model a *submodular field grammar* (SFG). Finding the MAP labeling to split a region into arbitrarily-shaped subregions is now tractable and we exploit this to develop an efficient approximate algorithm for MAP parsing of images with SFGs. Our algorithm, SFG-PARSE, is an iterative move-making algorithm that provably converges to a local minimum of the energy and reduces to $\alpha$-expansion for trivial grammars. Like other move-making algorithms, each step of SFG-PARSE chooses the best move from an exponentially large set of neighbors, thus overcoming many of the main issues with local minima [13]. Empirically, we compare SFG-PARSE to belief propagation and $\alpha$-expansion. We show that SFG-PARSE parses images in exponentially less time than both of these while returning comparable minima. Using deep convolutional neural network features as inputs, we investigate the modeling capability of SFGs. We show promising improvements in semantic segmentation accuracy when using SFGs in place of standard MRFs and when compared to the neural network features on their own.

Like SFGs, associative hierarchical MRFs [14, 15] also define multi-level MRFs, but use precomputed segmentations to set the regions of the non-terminal variables and thus do not permit arbitrary image regions. Neural parsing methods [16, 17] are grammar-like models for scene understanding, but use precomputed superpixels and thus also do not permit arbitrary region shapes. Most relevant is the work of Kumar and Koller [6] and Delong et al. [7], who define tree-structured submodular cost functions and use iterative fusion-style graph-cut algorithms for inference, much like SFG-PARSE. SFGs can be seen as an extension of these works that interprets the labelings at each level as productions in a grammar and permits multiple different productions of each symbol, thus defining a directed-acyclic-graph (DAG) cost function. This allows SFGs to be exponentially more expressive than these models with only a low-order polynomial increase in inference complexity. In the simple case of a tree-structured grammar (i.e., a non-recursive grammar in which each symbol only appears in the body of at most one production), SFGs and SFG-PARSE reduce to these existing approaches albeit without the label costs of Delong et al. [7]; however, it should be possible to extend SFGs in a similar manner.

In order to clearly describe and motivate SFGs, we present them here in the context of image parsing. However, SFGs are a general and flexible model class that is applicable anywhere grammars or MRFs are used, including social network modeling and probabilistic knowledge bases.

## 2 Preliminaries

### 2.1 Submodular MRFs

A Markov random field (MRF) for scene understanding defines a probabilistic model $p(\mathbf{y}, \mathcal{I}) = \frac{1}{Z} \exp(-E(\mathbf{y}, \mathcal{I}))$ over labeling $\mathbf{y} \in \mathcal{Y}^n$ and image $\mathcal{I}$, where $n = |\mathcal{I}|$ is the number of pixels, $Z = \sum_{\mathbf{y}' \in \mathcal{Y}^n} \exp(-E(\mathbf{y}', \mathcal{I}))$ is the partition function, and $\mathcal{Y}$ is the set of labels, which encode semantic classes such as Sky or Ground. MRFs for computer vision typically use pairwise energies $E(\mathbf{y}, \mathcal{I}) = \sum_{p \in \mathcal{I}} \theta_p(y_p, o_p) + \sum_{(p,q) \in \mathcal{I}} \theta_{pq}(y_p, y_q)$, where $\mathbf{y} = (y_0, \ldots, y_n)$ is a vector of labels; $o_p$ is the intensity value of pixel $p$; $\theta_p$ and $\theta_{pq}$ are the unary and pairwise energy terms for pixels $p$ and edges $(p, q)$, respectively; and, with a slight abuse of notation, we say that $\mathcal{I}$ contains both the nodes and edges in the MRF over the image. For binary labels $\mathcal{Y} = \{Y_1, Y_2\}$, an MRF is submodular if its energy satisfies $\theta_{pq}(Y_1, Y_1) + \theta_{pq}(Y_2, Y_2) \leq \theta_{pq}(Y_1, Y_2) + \theta_{pq}(Y_2, Y_1)$ for all edges $(p, q) \in \mathcal{I}$. If the energy is submodular, the MAP labeling $\mathbf{y}^* = \arg\max_{\mathbf{y} \in \mathcal{Y}^n} p(\mathbf{y}, \mathcal{I})$ can be computed exactly with a single graph cut in time $c(n)$, where $c(n)$ is worst-case low-order polynomial (the true complexity depends on the chosen min-cut/max-flow algorithm), but nearly linear time in practice [12, 13]. Thus, submodularity reduces the complexity of an optimization over $2^n$ states to nearly-linear time. While submodularity is useful for MAP inference, it also captures the fact that neighboring pixels in natural images tend to have the same label (e.g., Sky pixels appear next to other Sky pixels), which means that the MAP labeling in general partitions the image into contiguous regions of each label.

### 2.2 Image grammars

A context-free grammar (CFG) is a tuple $G = (N, \Sigma, R, S)$ containing a finite set of nonterminal symbols $N$; a finite set of terminal symbols $\Sigma$; a finite set of productions $R = \{v : X \rightarrow Y_1 \ldots Y_k\}$

with head symbol $X \in N$ and subconstituent symbols $Y_i \in N \cup \Sigma$ for $i = 1 \ldots k$; and a special start symbol $S \in N$ that does not appear on the right-hand side of any production. For scene understanding, a grammar for outdoor scenes might contain a production $S \rightarrow Sky\ Ground$, which would partition the image into Sky and Ground subregions.

To extend CFGs to images, we introduce the notion of a region $\mathcal{R} \subseteq \mathcal{I}$, which specifies a subset of the pixels and can have arbitrary shape. A parse (tree) $t \in \mathcal{T}_G(\mathcal{I})$ of image $\mathcal{I}$ with respect to grammar $G$ is a tree of nodes $n = (v, \mathcal{R})$, each containing a production $v \in R$ and a corresponding image region $\mathcal{R} \subseteq \mathcal{I}$, where $\mathcal{T}_G(\mathcal{I})$ is the set of valid parse trees for $\mathcal{I}$ under $G$, which we will write as $\mathcal{T}$ to simplify notation. For each node $n = (v, \mathcal{R})$ in a parse tree, the regions of its children $\{c_i = (v_i, \mathcal{R}_i) : c_i \in \mathrm{ch}(n)\}$ partition (segment) their parent's region such that $\mathcal{R} = \cup_i \mathcal{R}_i$ and $\cap_i \mathcal{R}_i = \varnothing$. If we let $v = X \rightarrow Y_1 \ldots Y_k$, then this partition is equivalently defined by a labeling $\mathbf{y}^v \in \mathcal{Y}_v^{|\mathcal{R}|}$ where $\mathcal{Y}_v = \{Y_1, \ldots, Y_k\}$, as there is a one-to-one correspondence between labelings and partitions of $\mathcal{R}$. Given a labeling for a production, the region of a subconstituent is simply the subset of pixels labeled as that subconstituent $\mathcal{R}_i = \{p : y_p^v = Y_i\}$ for any $i \in \{1, \ldots, k\}$.

A stochastic image grammar defines a generative probabilistic model of images by associating with each nonterminal a categorical distribution over the productions of that nonterminal. The generative process samples a production of the current nonterminal from this distribution, starting with the start symbol $S$ with the entire image as its region, and then partitions the current region into disjoint subregions – one for each subconstituent of the production. This process then recurses on each subconstituent-subregion pair, and terminates when a terminal symbol is produced, at which point the pixels for that region are generated. Formally, the probability of a parse $t \in \mathcal{T}$ of an image is $p(t, \mathcal{I}) = \prod_{(v,\mathcal{R}) \in t} p(v|\mathrm{head}(v)) \cdot p(\mathbf{y}^v|v, \mathcal{R})$, where $p(\mathbf{y}^v|v, \mathcal{R})$ specifies the probability of each labeling $\mathbf{y}^v \in \mathcal{Y}_v^{|\mathcal{R}|}$ (i.e., partition) of $\mathcal{R}$. Note that the above distribution over productions is the same categorical distribution as that used in PCFGs for natural language [18], but the distribution over segmentations is assumed to be uniform in PCFGs for natural language and is typically not made explicit. It is this latter distribution that causes representational challenges, as we must now specify a distribution $p(\mathbf{y}^v|v, \mathcal{R})$ for each production and for each of the $2^n$ possible image regions. We show how this can be achieved efficiently in the following section.

## 3 Submodular Field Grammars

As the main contribution of this work, we define (submodular) field grammars by combining the image grammars defined above with (submodular) MRFs. We do this by defining for each production $v$ an associated MRF over the full image $E_v(\mathbf{y}^v, \mathcal{I}) = \sum_{p \in \mathcal{I}} \theta_p^v(y_p^v) + \sum_{(p,q) \in \mathcal{I}} \theta_{pq}^v(y_p^v, y_q^v)$. A copy of this MRF is instantiated each time an instance (equivalently, a token, as this relates to the well-known type-token distinction) of $X$ is parsed as $v$, in the same way that each instance of a symbol in a grammar uses the same categorical distribution to select productions. In particular, an instance of a symbol has an associated region $\mathcal{R} \subseteq \mathcal{I}$ and the MRF instantiated for that instance is simply the subset of the full-image MRF containing all of the nodes in $\mathcal{R}$ and all of the edges between the nodes in $\mathcal{R}$. The energy of this instance is $E_v(\mathbf{y}^v, \mathcal{R}) = \sum_{p \in \mathcal{R}} \theta_p^v(y_p^v) + \sum_{(p,q) \in \mathcal{R}} \theta_{pq}^v(y_p^v, y_q^v)$. We thus write the labeling distribution as $p(\mathbf{y}^v|v, \mathcal{R}) \propto \exp(-E_v(\mathbf{y}^v, \mathcal{R}))$ and we write the energy of a parse tree (where each node contains production instances) as $E(t, \mathcal{I}) = \sum_{(v,\mathcal{R}) \in t} w_v + E_v(\mathbf{y}^v, \mathcal{R})$, where the weights $\{w_v\}$ parameterize each symbol's categorical distribution over productions and the probability of a parse tree is $p(t, \mathcal{I}) \propto \exp(-E(t, \mathcal{I}))$. To simplify notation, we will omit $v, \mathcal{I}$, and $\mathcal{R}$ when clear from context and sum over just $v$.

We refer to this model as a field grammar $G = (N, \Sigma, R, S, \Theta)$ parameterized by $\Theta$, which contains both the categorical weights and the MRF parameters. As in the image grammar formulation above, the pixels are generated when a terminal symbol is produced. Conversely, when parsing a given image, the unary terms $\{\theta_p^v\}$ can depend directly on the pixels of the image being parsed or on features of the image, as in a conditional random field. In our experiments, however, only the unary terms of the terminal symbols depend on the pixel values.

The MRFs in a field grammar can be parameterized arbitrarily but, in order to permit efficient MAP inference, we require that each term $\theta_{pq}^v$ satisfy the previously-stated binary submodularity condition for all edges $(p, q)$ and all productions $v : X \rightarrow Y_1 Y_2$ once the grammar has been converted to one in which each production has only two subconstituents, which is always possible and in the worst case increases the grammar size quadratically [18]. Note that it is easy to extend this to the non-binary case by requiring that the pairwise terms satisfy the $\alpha$-expansion or $\alpha\beta$-swap conditions [13], for example,

but we focus on the binary case here for simplicity. We also require that $\theta_{pq}^v(y_p^v, y_q^v) \geq \theta_{pq}^c(y_p^c, y_q^c)$ for every production $v \in R$, for every production $c$ that is a descendant of $v$ in the grammar, and for all possible labelings $(y_p^v, y_q^v, y_p^c, y_q^c)$, where $y_p^v, y_q^v \in \mathcal{Y}_v$ and $y_p^c, y_q^c \in \mathcal{Y}_c$. This ensures that segmentations of higher-level productions are submodular relative to their descendants, and captures a natural property of composition in images: that objects have larger regions than their parts. This means that the ratio of boundary length to region area is smaller for a symbol relative to its descendants, and thus its pairwise terms should be stronger. A grammar that satisfies these conditions is a *submodular field grammar* (SFG). Figure 1 shows a partial example of a (submodular) field grammar applied to image parsing, demonstrating the interleaved choices of productions and labelings, and the subregion decompositions resulting from these choices.

### 3.1 Relationship to other models

Above, we defined an SFG as an image grammar with an MRF at each production. An SFG can be equivalently reformulated as a planar MRF with one label for each path in the grammar. The number of such paths is exponential in the height of the grammar. This reformulation can be seen as follows. A parse tree over a region $\mathcal{R}$ has energy $E(t, \mathcal{R}) = \sum_{v \in t} w_v + E_v(\mathbf{y}^v, \mathcal{R}_v)$. We can rewrite this as $E(t, \mathcal{R}) = w(t) + \sum_{p \in \mathcal{R}} \theta_p^t + \sum_{(p,q) \in \mathcal{R}} \theta_{pq}^t$, where $w(t) = \sum_{v \in t} w_v$, $1[\cdot]$ is the indicator function, $\theta_p^t = \sum_{v \in t} \theta_p^v(y_p^v) \cdot 1[p \in \mathcal{R}_v]$, and $\theta_{pq}^t = \sum_{v \in t} \theta_{pq}^v(y_p^v, y_q^v) \cdot 1[(p,q) \in \mathcal{R}_v]$. This describes a flat MRF in which $\theta_p^t$ and $\theta_{pq}^t$ are the unary and pairwise terms. Inference in this flat MRF is not easier, and is likely harder, because it requires an exponentially-large set of labels and the hard constraints of the grammar must be enforced explicitly. However, this formulation will prove useful for our parsing algorithm.

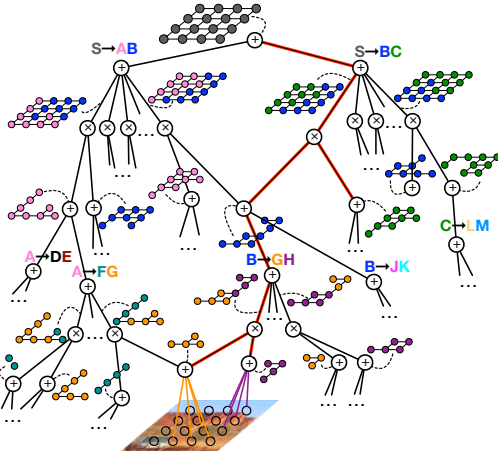

Figure 1: A DAG representing some of the possible production and labeling choices when parsing an image with an SFG. Each sum node represents either a choice of production for a particular region or a choice of labeling for the MRF representing a particular production of a region. Product nodes denote the partition of a region as defined by its labeling, where an MRF node's color denotes its label. Red edges denote a partial parse tree for the image shown at the bottom. Best viewed in color.

Another key benefit of our grammar-based formulation is sub-parse reuse, which enables exponential reductions in inference complexity and better sample complexity. For example, consider reusing a Wheel symbol among many vehicle types. Instead of having to learn and perform inference for each Wheel symbol (once per vehicle type and per vehicle-parent type, etc.), only one Wheel need be learned and inference on it performed only once.

Beyond PCFGs and MRFs, SFGs also generalize sum-product networks (SPNs) [19, 2]. Details on this mapping are given in the supplement. [1] Figure 1 shows a partial mapping of an SFG to an SPN.

## 4 Inference

When trying to understand natural scenes, it is important to recognize and reason about the relationships between objects. These relationships can be identified by finding the MAP parse of an image with respect to a grammar that encodes them, such as a submodular field grammar. The flat semantic labels traditionally used in scene understanding can also be recovered from this parse if they are encoded in the grammar, e.g., as the terminal symbols. We exploit this ability in our experiments.

For natural language, the optimal parse of a PCFG can be recovered exactly in time cubic in the length of the sentence with the CYK algorithm [20], which uses dynamic programming to efficiently parse a sentence in a bottom-up pass through the grammar. This is possible because each sentence only has a linear number of split points, meaning that all sub-spans of the sentence can be efficiently represented and enumerated. The key operation in the CYK algorithm is to compute the optimal parse of a given span $s$ (i.e., contiguous sub-sentence) as a given production $v : X \to YZ$ by explicitly iterating over all split points $i$ of that span, computing the probability of parsing $s$ as $v$ with split $i$,

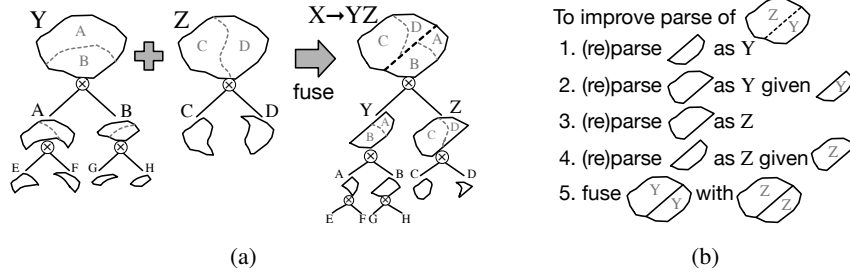

(a)                             (b)

Figure 2: The main components of SFG-PARSE: (a) Parsing a region $\mathcal{R}$ as $X \to YZ$ by fusing a parse of $\mathcal{R}$ as $Y \to AB$ with a parse of $\mathcal{R}$ as $Z \to CD$, and (b) Subsequently improving the parse of $\mathcal{R}$ as $X \to YZ$ by independently (re)parsing each of its subregions and then fusing these new parses. See text for more detail.

and choosing the split point with highest probability. The probability of parsing $s$ as $v$ with split $i$ is defined recursively as the product of $p(v|\text{head}(v))$ and the respective probabilities of the optimal parses of the two sub-spans as $Y$ and $Z$, respectively. CYK uses dynamic programming to cache the optimal parse of each sub-span as each symbol to avoid re-computing these unnecessarily.

Unfortunately, CYK applied to images is intractable because it is infeasible to enumerate all subregions of an image. Instead, we propose to construct (and cache) a parse of the entire image as each production and then use subregions of this parse to define the parse of each subregion, mirroring how distributions over subregions are defined in SFGs. We then exploit submodularity to find a locally optimal parse from an exponentially large set, without enumerating all such parses. Specifically, we optimally combine the parses of the subconstituents of a production to create a parse as that production. We refer to this procedure as *fusion* as it is analogous to the fusion moves of Lempitsky et al. [21].

### 4.1 Parse tree construction

Following Lempitsky et al. [21], let $\mathbf{y}^0, \mathbf{y}^1 \in \mathcal{Y}^n$ be two labelings of a submodular MRF with energy $E(\mathbf{y}, \mathcal{R}) = \sum_p \theta_p(y_p) + \sum_{pq} \theta_{pq}(y_p, y_q)$ and let $C(\mathbf{y}^0, \mathbf{y}^1) = \{\mathbf{y}^c\}$ denote the set of combinations of $\mathbf{y}^0$ and $\mathbf{y}^1$. A labeling $\mathbf{y}^c = (y_0^c, \ldots, y_n^c)$ is a combination of $\mathbf{y}^0 = (y_0^0, \ldots, y_n^0)$ and $\mathbf{y}^1 = (y_0^1, \ldots, y_n^1)$ iff each label in $\mathbf{y}^c$ is taken either from $\mathbf{y}^0$ or $\mathbf{y}^1$ such that $y_p^c \in \{y_p^0, y_p^1\}$ for all pixels $p = 1 \ldots n$. The fusion $\mathbf{y}^*$ of $\mathbf{y}^0$ and $\mathbf{y}^1$ is then defined as the minimum energy combination $\mathbf{y}^* = \arg\min_{\mathbf{y} \in C(\mathbf{y}^0, \mathbf{y}^1)} E(\mathbf{y}, \mathcal{R})$. Under certain conditions on $E$, fusion is a submodular minimization.

Recall that each parse tree $t$ equivalently corresponds to a particular labeling of a planar MRF with one label per path in the grammar. With a slight abuse of notation, we use $t$ to represent both the full parse tree and the corresponding planar MRF labeling. Let $v : X \to Y_1 Y_2$ be a production and $t_1, t_2$ be parses of some region $\mathcal{R} \subseteq \mathcal{I}$ as productions $u_1 : Y_1 \to Z_1 Z_2$ and $u_2 : Y_2 \to Z_3 Z_4$, respectively.

**Definition 1.** *For production $v : X \to Y_1 Y_2$ and parse trees $t_1, t_2$ over region $\mathcal{R}$ with head symbols $Y_1, Y_2$, the* fusion *of $t_1$ and $t_2$ as $v$ is the minimum energy parse tree $t_v = \arg\min_{t \in C(t_1, t_2)} E(t, \mathcal{R})$ constructed from the combination of $t_1$ and $t_2$, with $(v, \mathcal{R})$ appended as root.*

Because $t_1$ and $t_2$ are MRF labelings, we can fuse them to create a new parse tree $t_v$ in which each pixel in $\mathcal{R}$ is labeled with the same path that it had in either $t_1$ or $t_2$. When we do this, we prepend $v$ to each pixel's label, which is equivalent to adding $(v, \mathcal{R})$ as the new root node of $t_v$. Figure 2a shows an example of fusing two parse trees to create a new parse tree.

**Proposition 1.** *The fusion of two parse trees is a submodular minimization.*

Although fusion requires finding the optimal labeling from an exponentially large set, two parse trees can be fused with a single graph cut by exploiting submodularity. Proofs are given in the supplement.

Finally, we define the union of two parse trees $t = t_1 \cup t_2$ that have the same productions but are over disjoint regions (i.e., $\mathcal{R}_1 \cap \mathcal{R}_2 = \varnothing$) as the parse tree $t$ in which the region of each node in $t$ is the union of the regions of the corresponding nodes in $t_1$ and $t_2$.

### 4.2 SFG-Parse

Pseudocode for our parsing algorithm, SFG-PARSE, is presented in Algorithm 1. SFG-PARSE is an iterative move-making algorithm that efficiently and provably converges to a local minimum of the energy function. Currently, SFG-PARSE applies only to non-recursive grammars, but we believe it will be straightforward to extend it to recursive ones.

To parse an image with respect to a given non-recursive grammar, SFG-PARSE starts at the terminal

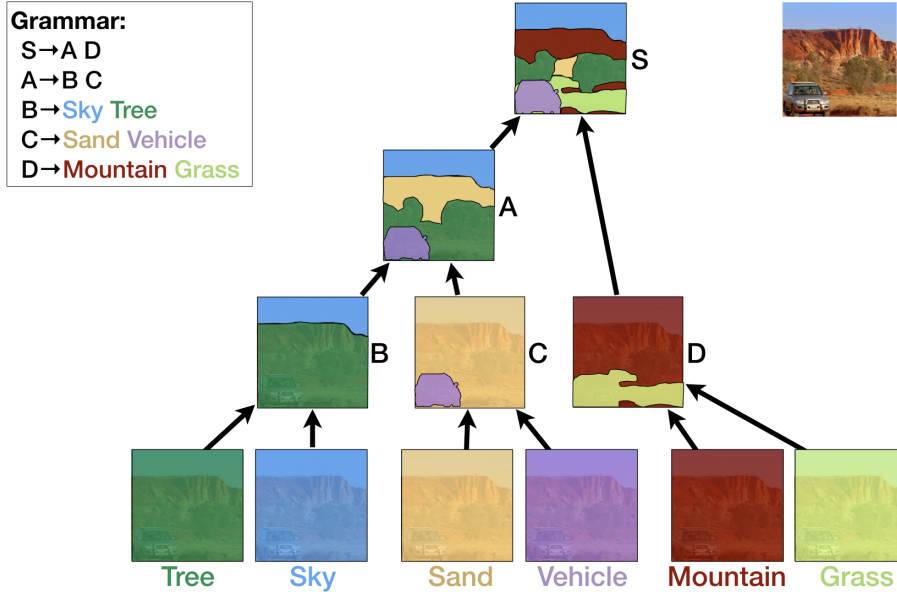

**Grammar:**
S→A D
A→B C
B→Sky Tree
C→Sand Vehicle
D→Mountain Grass

Tree    Sky    Sand    Vehicle    Mountain    Grass

Figure 3: One iteration of SFG-PARSE applied to the image shown on the right with respect to the simple grammar on the left. Proceeding from bottom to top, SFG-PARSE first parses the image as each of the terminal symbols (i.e., each pixel in the image is labeled as that terminal symbol), and then fuses these to create parses of the image as symbols $B$, $C$, and $D$. These parses are then fused in turn to create parses of the image as $A$ and finally $S$. The final full parse tree returned is the parse of $S$.

symbols and moves backwards through the productions towards the start symbol (line 9), constructing and caching a parse of the full image as each production. The parse for each production is constructed by fusing the cached parses of that production's subconstituents (lines 13 and 14). An example of this procedure is shown in Figure 3, where the parses of symbols $S$, $A$, $B$, $C$, and $D$ are constructed by fusing the parses of the subconstituents of their respective productions. For simplicity, the grammar in Figure 3 only contains a single production for each symbol, and no symbol is a subconstituent of multiple productions; in general, however, most symbols in the grammar appear on both the left- and right-hand sides of multiple productions. To accommodate this, SFG-PARSE maintains multiple instances (aka. tokens) of each symbol and chooses the appropriate production and instance during parsing. This is discussed in more detail below. Subsequent iterations of SFG-PARSE simply repeat this bottom-up procedure while ensuring that for each (re)parse of a production, the previous iteration's parse of that production (or a guaranteed lower energy alternative) can be constructed via fusion. This guarantees convergence of SFG-PARSE.

In CYK, each span of a sentence is explicitly parsed as each production, making it straightforward to have multiple instances of a symbol. However, since there are an exponential number of subregions of an image, SFG-PARSE instead constructs a parse of the entire image for each production and reuses that parse for each of these subregions. To ensure consistency of this parse, if only one parse were allowed per production then each instance would have to be parsed with the exact same set of productions, a severe restriction on the expressivity of the model. To avoid this, SFG-PARSE permits multiple instances of a symbol $X$, one per unique path from the root to a production of $X$ in $\hat{t}$, where $\hat{t}$ is the best parse of $S$ from the previous iteration. This allows the number of instances of each symbol to grow or shrink at each iteration according to the parse constructed in the previous iteration. Processing of instances and their corresponding regions occurs on lines 5-6. For each instance $x$ of symbol $X$ in $\hat{t}$ for a production $v : X \to YZ$, SFG-PARSE records pointers to $x$'s child instances $y$ and $z$, which are later (line 12) used to determine which instances of $Y$ and $Z$ to fuse when parsing $v$.

In the common scenario that a symbol has no instances – either because it doesn't appear in $\hat{t}$ or because $\hat{t}$ was not provided – then that symbol is assigned the region containing the entire image as an instance (line 7), which serves as a powerful initialization method. If a symbol has no instances, then it did not appear in $\hat{t}$ and its parse can be constructed by fusing any instances of its production's subconstituents without affecting convergence. In the rare case that a symbol has multiple instances, one can be chosen either by estimating a bound on the energy or even randomly (line 12).

---

**Algorithm 1** Compute the (approximate) MAP parse of an image with respect to an SFG.

---

**Input:** The image $\mathcal{I}$, a non-recursive SFG $G = (N, \Sigma, R, S, \Theta)$, and an (optional) input parse $\hat{t}$.
**Output:** A parse of the image, $t^*$, with energy $E(t^*, \mathcal{I}) \leq E(\hat{t}, \mathcal{I})$.

  1: **function** SFG-PARSE($\mathcal{I}, G, \hat{t}$)
  2:     **for** each terminal $T \in \Sigma$ **do** $t_{\mathcal{R}_T} \leftarrow$ the trivial parse with all pixels parsed as $T$
  3:     **while** the energy of any production of the start symbol $S$ has not converged **do**
  4:         *// record the instances (i.e., regions) of each symbol in $\hat{t}$ and initialize instance-less symbols*
  5:         **for** each node in $\hat{t}$ with production $u : X \rightarrow YZ$, region $\mathcal{R}_X$, and subregions $\mathcal{R}_Y, \mathcal{R}_Z$ **do**
  6:             append $\mathcal{R}_Y, \mathcal{R}_Z$ to region lists $\mathcal{R}[Y], \mathcal{R}[Z]$ and set as the child regions of $u$ for $\mathcal{R}_X$
  7:         **for** each symbol $X \in N$ with no regions in $\mathcal{R}[X]$ **do** append $\mathcal{R}_X = \mathcal{I}$ to $\mathcal{R}[X]$
  8:         *// perform the upward pass to parse with the SFG at this iteration*
  9:         **for** each symbol $X \in N$, in reverse topological order **do**
10:             **for** each region $\mathcal{R}_X$ in region list $\mathcal{R}[X]$ **do**     *// each region is an instance (token) of $X$*
11:                 **for** each production $v : X \rightarrow YZ$ of symbol $X$ **do**
12:                     $\mathcal{R}_Y, \mathcal{R}_Z \leftarrow$ the child regions of $v$ for $\mathcal{R}_X$ if they exist, else choose heuristically
13:                     $t_v, e_v \leftarrow$ fuse $t_{\mathcal{R}_Y}$ and $t_{\mathcal{R}_Z}$ as production $v$ over region $\mathcal{R}_X$
14:                     $t_{\bar{v}}, e_{\bar{v}} \leftarrow$ fuse $t_{\mathcal{R}_Y}$ and $t_{\mathcal{R}_Z}$ as production $v$ over region $\mathcal{R}_{\overline{X}} = \mathcal{I} \backslash \mathcal{R}_X$ given $t_v$
15:                 $t_{\mathcal{R}_X}, e_{\mathcal{R}_X} \leftarrow$ the full parse $t_v \cup t_{\bar{v}}$ with lowest energy $e_v$ *// choose best parse of $\mathcal{R}_X$*
16:         $\hat{t}, \hat{e} \leftarrow t_{\mathcal{R}_S}, e_{\mathcal{R}_S}$         *// $S$ only ever has a single region, which contains all of the pixels*
17:     **return** $\hat{t}, \hat{e}$

---

If a symbol $X$ does have an instance in $\hat{t}$, SFG-PARSE first parses only that instance's region $\mathcal{R}_X$ into tree $t_v$ (line 13) and then parses the remainder of the image $\mathcal{R}_{\overline{X}}$ as $v$ given the partial parse $t_v$ (line 14). The union of these gives a full parse of the entire image as $v$ for this instance. Parsing an instance in two parts is necessary to ensure that SFG-PARSE never returns a worse parse. Figure 2b shows an inefficient version of the process for re-parsing an instance of $X$, where first the subregions labeled as $Y$ and $Z$ are re-parsed (steps 1-2), then the remaining pixels are re-parsed given the other parses (steps 3-4), and finally the unions of these parses are fused to get a parse of the region as $X$ (step 5). For efficiency reasons, SFG-PARSE does not actually reparse $Y$ and $Z$ for each production that produces them; instead, their parses are cached and re-used. We define parsing a region $\mathcal{R}_{\overline{X}}$ given a parse $t_v$ of another region $\mathcal{R}_X$ to mean that each pairwise term with a pixel in each region already has the label of the pixel in $\mathcal{R}_X$ set to its value in $t_v$ (i.e., like conditioning in a probabilistic model).

Finally, the parse of the production $u$ with the lowest energy over $\mathcal{R}_X$ is then chosen as its parse (line 15). At the end of the upward pass, the parse of the full image $\hat{t}$ is simply the parse of the start symbol's region, which always contains all pixels (line 16).

### 4.3 Analysis

In this section, we analyze the convergence and computational complexity of SFG-PARSE.

**Theorem 1.** *Given a parse $\hat{t}$ of $S$ over the entire image with energy $E(\hat{t})$, each iteration of* SFG-PARSE *constructs a parse $t$ of $S$ over the entire image with energy $E(t) \leq E(\hat{t})$ and, since the minimum energy of an image parse is finite,* SFG-PARSE *will always converge.*

Theorem 1 shows that SFG-PARSE always converges to a local minimum of the energy function. Like other move-making algorithms, SFG-PARSE explores an exponentially large set of moves at each step, so the returned local minimum is generally much better than those returned by more local procedures [13]. Further, we typically observe convergence in fewer than ten iterations, with the majority of the energy improvement occurring in the first iteration.

**Proposition 2.** *Let $c(n)$ be the time complexity of computing a graph cut on $n$ pixels and $|G|$ be the size of the grammar defining the SFG. Then each iteration of* SFG-PARSE *takes time $O(|G|c(n)n)$.*

Proposition 2 shows that each iteration of SFG-PARSE has complexity $O(|G|c(n)n)$, where $n$ is the number of pixels and $c(n)$ is the complexity of the graph-cut algorithm, which is low-order polynomial in $n$ in the worst case, but nearly linear-time in practice [12, 13]. The additional factor of $n$ is due to the number of regions (i.e., instances) of each symbol, which in the worst case is $O(n)$ but in practice is almost always a small constant (often one). Thus, SFG-PARSE typically runs in time $O(|G|c(n))$.

Note that directly applying $\alpha$-expansion to parsing an SFG requires optimizing an MRF with one

label for each path in the grammar, which would take time exponential in the height of the grammar.

SFG-PARSE can be extended to productions with more than two subconstituents by replacing the internal graph cut used to fuse subtrees with a multi-label algorithm such as $\alpha$-expansion. SFG-PARSE would still converge because each subtree would still never increase in energy. Alternatively, an algorithm such as QPBO [22] could be used, which would obviate the submodularity requirement.

## 5    Experiments

We evaluated our model and inference algorithm in two experiments, both using unary features from DeepLab [23, 24], a state-of-the-art convolutional semantic segmentation network. First, to evaluate the performance of SFG-PARSE, we programmatically generated SFGs and compared the runtime of and minimum energy returned by SFG-PARSE to that of $\alpha$-expansion and max-product belief propagation (BP), two standard MRF inference algorithms. Second, to evaluate SFGs as a model of natural scenes, we segmented images at multiple levels of granularity, used these segmentations to generate SFGs over the DeepLab features (in place of the raw pixel intensities), and compared the segmentation accuracy resulting from parsing the generated SFGs using SFG-PARSE to that of using (a) DeepLab features alone and (b) a planar submodular MRF on the DeepLab features.

The DeepLab features are trained on the Stanford Background Dataset (SBD) [5] training set. Evaluations are performed on images from the SBD test set. In all MRFs (planar and in each SFG), the pairwise terms are standard contrast-dependent boundary terms [25] multiplied by a single weight, $w_{\mathrm{BF}}$.

### 5.1    Inference evaluation

To evaluate the performance of SFG-PARSE, we programmatically generated SFGs while varying their height, number of productions per nonterminal (#prods), and strength of the pairwise (boundary) terms. Each algorithm was evaluated using the same grammars, DeepLab features, and randomly selected images. We compared the performance of SFG-PARSE to that of running $\alpha$-expansion on a flat pairwise MRF containing one label for each possible parse path in the grammar and also to running BP on a multi-level (3-D) pairwise MRF with the same height as the grammar. These are the natural comparisons, as existing hierarchical MRF algorithms do not support the DAG structure that makes SFGs so powerful. Details of these models and additional figures are provided in the supplement.

Increasing boundary strength, grammar height, and #prods each make inference more challenging. Individual pixels cannot be flipped easily with stronger boundary terms, while grammar height and #prods both determine the number of paths in the grammar. Figure 4a plots the average minimum energy of the parses found by each algorithm versus the boundary factor, $w_{\mathrm{BF}}$ (x-axis is log scale) and Figure 4d plots inference time versus boundary factor. As shown, SFG-PARSE returns comparable or better parses to both BP and $\alpha$-expansion and in less time. In Figure 4e, we set $w_{\mathrm{BF}}$ to 20 and plot inference time versus grammar height. The energies are shown in Figure 4b. As expected, inference time for SFG-PARSE scales linearly with height, whereas it scales exponentially for both $\alpha$-expansion and BP. Again, the energies and accuracies of the parses returned by SFG-PARSE are nearly identical to those of $\alpha$-expansion. Finally, we set $w_{\mathrm{BF}}$ to 20 and plot inference time versus #prods in Figure 4f, and energy versus #prods in Figure 4c. Once again, SFG-PARSE returns equivalent parses to $\alpha$-expansion and BP but in much less time.

### 5.2    Model evaluation

To evaluate whether natural scenes exhibit the compositional part-subpart structure over arbitrarily-shaped regions that SFGs can capture but previous methods cannot, we generated grammars on SBD images where the semantic labels were the terminals. We then computed the mean pixel accuracy of the terminal labeling from the parse tree returned by SFG-PARSE.

Grammars were generated (not learned) as follows. We first over-segmented each of the 143 test images at 4 different levels of granularity and intersected the most fine-grained of these with the label regions. We created a unique grammar for each image by taking that image's over-segmentations and the over-segmentations of four other randomly chosen images and adding a symbol for each contiguous region in each segmentation. We then added productions between overlapping segments for each subsequent pair of granularity levels within each image and across images. Finally, we added terminal productions from the symbols in the most granular level, where each terminal production can produce only those labels that occur in its head symbol's corresponding segment (note that we similarly restricted the possible labels produced by other models to ensure the comparison was fair). On average, each induced grammar had 860 symbols and 1250 productions with 5 subconstituents each. The features output by DeepLab were used as the unaries in the MRFs of the terminal

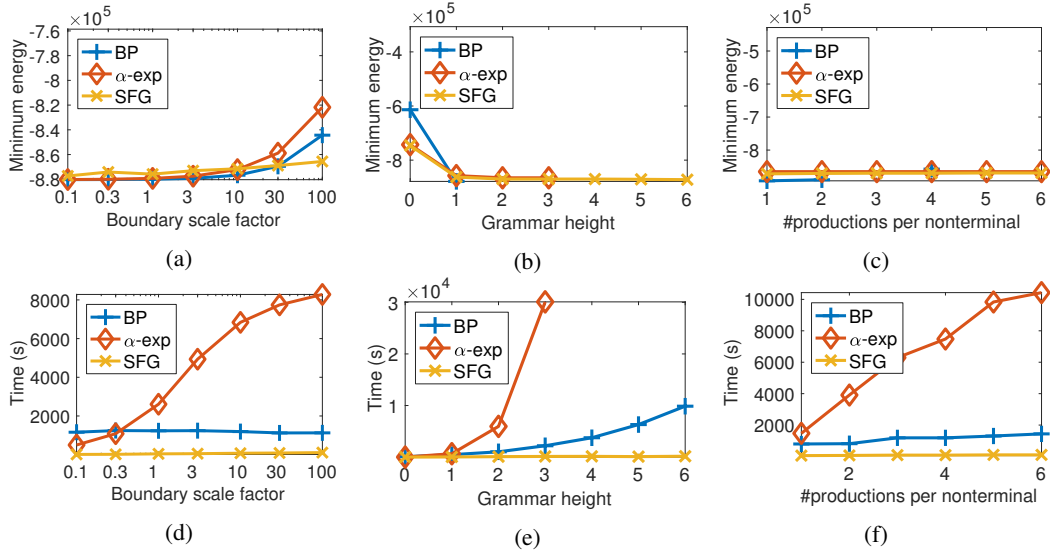

Figure 4: The energy of the returned parse (a,b,c) and total running time (d,e,f) when evaluating MAP inference using belief propagation, $\alpha$-expansion, and SFG-PARSE while varying (a,d) boundary strength, (b,e) grammar height, and (c,f) number of productions. In all figures, lower is better. Each data point is the average over the same 10 randomly-selected images. Missing data points for BP indicate that it returned an inconsistent parse with infinite energy. Missing data points for $\alpha$-expansion indicate that it ran out of time or memory. Figures S1, S2, and S3 in the supplement show the mean pixel accuracies for each experiment.

productions. All productions had uniform probability and the same MRF parameters were used across all images. This ensured that any improvement in performance was due solely to the structure of the underlying grammar. Further details about the induced grammars are provided in the supplement.

After parsing each image with respect to its grammar, we computed the mean pixel accuracy of the terminal labeling of the parse. We compared this to the accuracy of the DeepLab features alone and to the accuracy of a standard flat submodular MRF over the DeepLab features, with pairwise terms set in the same way as in the SFGs.

| DeepLab | DeepLab+MRF | DeepLab+SFG |
|---------|-------------|-------------|
| 87.77   | 87.93       | **90.03**   |

Table 1: Mean pixel accuracy on 143 SBD test images.

These results are shown in Table 1, which shows a 20% relative decrease in error for SFGs, which is quite remarkable given how well the DeepLab features do on their own and how little the flat MRF helps. While this does not constitute a full evaluation of SFGs for semantic segmentation as we did not learn the SFGs, it provides evidence that SFGs are a compelling model class. In the supplement, we propose an approach for learning SFGs but we leave its implementation and evaluation for future work as it requires the creation of new datasets of parsed images, which is outside the scope of this paper. Even without learning, however, this experiment demonstrates that natural scenes do exhibit high-level compositional structure and that SFGs are able to efficiently exploit this structure to improve scene understanding and image parsing.

## 6 Conclusion

This paper proposed submodular field grammars (SFGs), a novel stochastic image grammar formulation that combines the expressivity of image grammars with the efficient combinatorial optimization capabilities of submodular MRFs. SFGs are the first image grammars to enable efficient parsing of objects with arbitrary region shapes. To achieve this, we presented SFG-PARSE, a move-making algorithm that exploits submodularity to find the (approximate) MAP parse of an SFG. Analytically, we showed that SFG-PARSE is both convergent and fast. Empirically, we showed (i) that SFG-PARSE achieves accuracies and energies comparable to $\alpha$-expansion – which returns optima within a constant factor of the global optimum – while taking exponentially less time to do so and (ii) that SFGs are able to represent the compositional structure of images to better parse and understand natural scenes.

In future work, we plan to focus on learning the parameters and structure of SFGs, as we believe that their unique combination of tractability and expressivity will lead to better understanding of natural scenes. We also plan to apply SFGs to other domains, such as activity recognition, social network modeling, and probabilistic knowledge bases.

**Acknowledgements**

AF would like to thank Robert Gens, Rahul Kidambi, and Gena Barnabee for useful discussions, insights, and assistance with this document. The DGX-1 used for this research was donated by NVIDIA. This research was partly funded by ONR grant N00014-16-1-2697 and AFRL contract FA8750-13-2-0019. The views and conclusions contained in this document are those of the authors and should not be interpreted as necessarily representing the official policies, either expressed or implied, of ONR, AFRL, or the United States Government.

## Footnotes

[1]Supplementary material is available at https://homes.cs.washington.edu/~pedrod/papers/neurips18sp.pdf.

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
