[Supplementary Material]

# Submodular Field Grammars: Representation, Inference, and Application to Image Parsing

## (Supplementary Material)

**Abram L. Friesen** and **Pedro Domingos**
Paul G. Allen School of Computer Science and Engineering
University of Washington
Seattle, WA 98195
{afriesen,pedrod}@cs.washington.edu

# 1 Additional model details

## 1.1 SFG to SPN conversion

Beyond PCFGs and MRFs, SFGs also generalize sum-product networks (SPNs) [1, 2]. Briefly, an SPN is a deep but tractable probabilistic model typically represented as a directed acyclic graph (DAG) of sum nodes, product nodes, and leaf functions. Product nodes in an SPN are decomposable, meaning that the children must have disjoint scopes, where a node's scope is the set of variables its descendant leaf functions are over.

An SFG defines an SPN containing a sum node for each possible region of each nonterminal, a product node for each segmentation of each production of each possible region of each nonterminal, and a leaf function on the pixels of the image for each possible region of the image for each terminal symbol. The children of the sum node $s$ for nonterminal $X_s$ with region $\mathcal{R}_s$ are all product nodes $r$ with a production $v_r : X_s \rightarrow Y_1 \ldots Y_k$ and region $\mathcal{R}_{v_r} = \mathcal{R}_s$. Each product node corresponds to a labeling $\mathbf{y}^{v_r}$ of $\mathcal{R}_{v_r}$ and the edge to its parent sum node has weight $\exp(-w_v - E(\mathbf{y}^{v_r}, \mathcal{R}_{v_r}))$. The children of product node $r$ are the sum or leaf nodes with matching regions that correspond to the constituent nonterminals or terminals of $v_r$, respectively. Since the SPN resulting from an SFG must represent each possible subregion of the image, this SPN is exponentially large and inference in it is intractable.

Note that this underlying SPN is decomposable, but not smooth. However, [3] showed that smoothness was not a necessary condition for tractable inference and that no corrective factor is necessary when operating in the min-sum semiring, which is what is used for finding the (approximate) optimal parse of an SFG.

A key benefit of SFGs in comparison to previous grammar-based approaches is that regions can have arbitrary shapes and are not restricted to a small class of shapes such as rectangles [2, 4]. This flexibility is important when parsing images, as real-world objects and abstractions can take any shape, but it comes with a combinatorial explosion of possible parses. However, by exploiting submodularity, we are able to develop an efficient inference algorithm for SFGs, allowing us to efficiently parse images into a hierarchy of arbitrarily-shaped regions and objects, yielding a very expressive model class. This efficiency is despite the size of the underlying SPN, which is in general far too large to explicitly instantiate.

## 2 Learning SFGs

In this section, we present an initial algorithm for learning SFGs but leave testing this approach for future work. An SFG is parameterized by its production costs $\mathbf{w}_s = \{w_v : v \in R\}$, which are the log-space version of an SPN's weights, and its MRF weights $\mathbf{w}_m$. The parameters of an SFG can be learned in a multitude of ways, but we propose here an approach that builds on the insight of Section 3.1 of the main paper: given the true region for each symbol, an SFG reduces to an SPN. Conversely, given the true production for each symbol, an SFG reduces to an MRF. We thus propose to learn SFG parameters using an alternating-minimization approach where the SPN weights $\mathbf{w}_s$ are first updated with the MRF parameters held fixed and then the MRF weights $\mathbf{w}_m$ are updated with the SPN weights held fixed, iterating until convergence. While it is possible to learn $\mathbf{w}_s$ and $\mathbf{w}_m$ simultaneously, our preliminary investigations indicate that learning them separately provides a more stable approach, where symbols are first associated with different image patches (or features) by updating $\mathbf{w}_s$, and then each symbol's region and appearance weights $\mathbf{w}_m$ are fit to that symbol's image patches (or features).

For updating the weights, both SPNs and MRFs can be learned both generatively and discriminatively, and SFGs are no different; we focus here on the discriminative case, but the generative case is similar. As with both SPNs and MRFs, the derivative of the conditional log-likelihood of an SFG with respect to a weight $w_i$ is simply the difference of the expected count of the corresponding production (or pixel or edge) over all parse trees that are compatible with both the labels and the image and the expected count over parse trees compatible with only the image; i.e.,

$$\frac{\partial}{\partial w_i} \log p_{\mathbf{w}}(\mathbf{y}|\mathcal{I}) = \mathbb{E}_{t \in \mathcal{T}_{\mathbf{w}}(\mathbf{y}, \mathcal{I})}[n_i(t)|\mathbf{y}, \mathcal{I}] - \mathbb{E}_{t' \in \mathcal{T}_{\mathbf{w}}(\mathcal{I})}[n_i(t')|\mathcal{I}], \tag{1}$$

where $\mathbf{y}$ are the query variables, $\mathcal{T}_{\mathbf{w}}(\mathbf{y}, \mathcal{I})$ and $\mathcal{T}_{\mathbf{w}}(\mathcal{I})$ are the sets of parse trees compatible with their respective arguments, and $n_i(t)$ is the count of weight $w_i$ in $t$. Unfortunately, since no datasets of parsed images exist to train on, both expectations are intractable. However, an effective approximation to the second expectation is to simply use the counts from the MAP parse, which is accurate if it has much of the probability mass; this is known as voted perceptron [5] and has been used to efficiently train both MRFs [6] and SPNs [7]. In SPNs, both expectations are tractable but are still replaced with their MAP state to overcome vanishing gradients. We propose to extend this method to SFGs, and approximate each expectation with the counts from its respective MAP parse, as found by SFG-PARSE. The gradient update is then simply

$$\frac{\partial}{\partial w_i} \log p_{\mathbf{w}}(\mathbf{y}|\mathcal{I}) \approx n_i(t^*_{\mathbf{y}\mathcal{I}}) - n_i(t^*_{\mathcal{I}}), \tag{2}$$

where $t^*_\cdot = \arg\min_{t \in \mathcal{T}_{\mathbf{w}}(\cdot)} E(t, \mathcal{I})$.

## 3 Proofs

**Proposition 1.** *The fusion of two parse trees is a submodular minimization.*

*Proof.* First, let every submodular MRF energy $E^u(\mathbf{y}^u, \mathcal{R})$ be in normal form such that $\theta^u_{pq}(y, y) = 0$ and $\theta^u_{pq}(y^u_p, y^u_q) \geq 0$ for all labels $y, y^u_p, y^u_q \in \mathcal{Y}_u$, where any submodular energy can be reparameterized into normal form in time linear in the region size [8].

Now, let $t_0, t_1$ be parse trees over region $\mathcal{R}$ with respective head symbols $Y_0, Y_1$ and let the fusion be for a production $v : X \to Y_0 Y_1$. The fusion of parse trees $t_0, t_1$ is given by $t^* = \arg\min_{t \in C(t_0, t_1)} E(t, \mathcal{R})$, where $E(t, \mathcal{R}) = \sum_{v \in t} w(t) + \sum_{p \in \mathcal{R}} \theta^t_p + \sum_{(p,q) \in \mathcal{R}} \theta^t_{pq}$ and $\theta^t_{pq} = \sum_{v \in t} \theta^v_{pq}(y^v_p, y^v_q)$. However, because each $\theta^v_{pq}$ is in normal form and any parse tree can contain only one production $c \in t$ in which $y^c_p \neq y^c_q$ for any neighboring pixels $p, q$ (because these pixels are placed in different regions in all descendants of the region in which they are labeled differently), it follows that the summation for $\theta^t_{pq}$ contains at most one non-zero term and thus $\theta^t_{pq} = \theta^c_{pq}(y^v_p, y^v_q)$.

Finally, by introducing an auxiliary binary variable $\mathbf{z} \in \{0, 1\}^{|\mathcal{R}|}$, we can define the combination of $t_0$ and $t_1$ as $t^c(\mathbf{z}) = t_0 \circ (1 - \mathbf{z}) + t_1 \circ (\mathbf{z})$, where $\circ$ is the Hadamard (element-wise) product. The fusion problem can now be written as a binary minimization problem $t^* = \arg\min_{\mathbf{z} \in \{0,1\}^{|\mathcal{R}|}} E(t(\mathbf{z}), \mathcal{R})$, where the energy is submodular if $\theta^{t(z_p=0, z_q=1)}_{pq} + \theta^{t(z_p=1, z_q=0)}_{pq} \geq \theta^{t_0}_{pq} + \theta^{t_1}_{pq}$. Because the pairwise

terms are in normal form we have that $\theta_{pq}^{t_0} = \theta_{pq}^{c_0}$ and $\theta_{pq}^{t_0} = \theta_{pq}^{c_1}$, where $c_0$ and $c_1$ are the productions in $t_0$ and $t_1$, respectively, at which pixels $p$ and $q$ were labeled differently (note that the values of their labels are unimportant here). Further, we have that $\theta_{pq}^{t(z_p=0, z_q=1)} = \theta_{pq}^v(Y_0, Y_1)$ and $\theta_{pq}^{t(z_p=1, z_q=0)} = \theta_{pq}^v(Y_1, Y_0)$ because by choosing them from different trees, the production $v$ that is being fused is now the production at which these pixels are first labeled differently. However, since from the definition of an SFG we have that $\theta_{pq}^v(y_p^v, y_q^v) \geq \theta_{pq}^c(y_p^c, y_q^c)$ for $c$ any possible descendant production of $v$ and for all labelings, then it follows that $\theta_{pq}^v(Y_0, Y_1) \geq \theta_{pq}^{c_0}$ and $\theta_{pq}^v(Y_1, Y_0) \geq \theta_{pq}^{c_1}$ and thus the submodularity condition holds.

$\square$

The following result shows how SFG-PARSE can improve a parse tree while ensuring that the energy of that parse tree never gets worse.

**Lemma 1.** *Given a labeling $\mathbf{y}^v$ which fuses parse trees $t_1, t_2$ into $t$ with root production $v$, energy $E(t, \mathcal{R})$, and subtree regions $\mathcal{R}_1 \cap \mathcal{R}_2 = \emptyset$ defined by $\mathbf{y}^v$, then any improvement $\Delta$ in $E(t_1, \mathcal{R}_1)$ also improves $E(t, \mathcal{R})$ by at least $\Delta$, regardless of any change in $E(t_1, \mathcal{R} \backslash \mathcal{R}_1)$.*

*Proof.* Since the optimal fusion can be found exactly, and the energy of the current labeling $\mathbf{y}^v$ has improved by $\Delta$, the optimal fusion will have improved by at least $\Delta$. $\square$

**Proposition 2.** *Let $c(n)$ be the time complexity of computing a graph cut on $n$ pixels and $|G|$ be the size of the grammar defining the SFG, then each iteration of SFG-PARSE takes time $O(|G|c(n)n)$.*

*Proof.* Let $k$ be the number of productions per nonterminal symbol and $N$ be the nonterminals. The three main loops of the algorithm have complexity $|N|$, $n$ (because there can be at most $n$ regions and the regions are disjoint), and $k$, respectively. For line 8, the choice of parses for productions in $\hat{t}$ takes constant time, and the rest can be chosen arbitrarily. For lines 9-10, the fusion of a region $\mathcal{R}$ has complexity $O(|\mathcal{R}| + c(|\mathcal{R}|)) = O(c(|\mathcal{R}|))$, so the worst-case complexity of the inner loop is when $\mathcal{R}$ is empty or the full image, giving complexity $O(c(n))$. Thus, the total complexity of each iteration of SFG-PARSE is $O(|N|k \cdot c(n) \cdot n) = O(|G|c(n)n)$. $\square$

**Theorem 1.** *Given a parse $\hat{t}$ of $S$ over the entire image with energy $E(\hat{t})$, each iteration of SFG-PARSE constructs a parse $t$ of $S$ over the entire image with energy $E(t) \leq E(\hat{t})$, and since the minimum energy of an image parse is finite, SFG-PARSE will always converge.*

*Proof.* We will prove by induction that for all nodes $n \in \hat{t}$ with corresponding production $v : X \to YZ$, region $\mathcal{R}_X$, subtree $\hat{t}_{\mathcal{R}_X}$ over region $\mathcal{R}_X$, and child subtrees $\hat{t}_{\mathcal{R}_Y}, \hat{t}_{\mathcal{R}_Z}$ over regions $\mathcal{R}_Y, \mathcal{R}_Z$, that $E(t_{\mathcal{R}_X}) \leq E(\hat{t}_{\mathcal{R}_X})$ after one iteration. Since the start symbol $S$ has only one region containing the entire image, this proves the claim.
**Base case.** Let $\hat{t}_{\mathcal{R}_X}$ be a subtree with region $\mathcal{R}_X$ and production $v : X \to Y$ containing only a single terminal child and let $\{u_i = X \to Y_i\}$ be the set of productions of $X$ (where such a $\hat{t}_{\mathcal{R}_X}$ must exist because the grammar is non-recursive and terminates). By definition, $t_v = \hat{t}_{\mathcal{R}_X}$, where $t_v$ is the new parse of $\mathcal{R}_X$ as $v$, because terminal parses do not change for the same region. Then, since $t_{\mathcal{R}_X} = \arg\min_{u_i} E(t_{u_i})$ and $v \in \{u_i\}$, it immediately follows that $E(t_{\mathcal{R}_X}) \leq E(\hat{t}_{\mathcal{R}_X})$ and the claim holds.
**Induction step.** Let $n \in \hat{t}$ be a node in $\hat{t}$ with corresponding production $v : X \to YZ$, region $\mathcal{R}_X$, subtree $\hat{t}_{\mathcal{R}_X}$ over region $\mathcal{R}_X$, and child subtrees $\hat{t}_{\mathcal{R}_Y}, \hat{t}_{\mathcal{R}_Z}$ over regions $\mathcal{R}_Y, \mathcal{R}_Z$, such that $\mathcal{R}_Y \cup \mathcal{R}_Z = \mathcal{R}_X$ and $\mathcal{R}_Y \cap \mathcal{R}_Z = \emptyset$, and suppose that $E(t_{\mathcal{R}_Y}) \leq E(\hat{t}_{\mathcal{R}_Y})$ and $E(t_{\mathcal{R}_Z}) \leq E(\hat{t}_{\mathcal{R}_Z})$. From Lemma 1, it follows that the parse $t_v$ computed from fusing $t_{\mathcal{R}_Y}$ and $t_{\mathcal{R}_Z}$ in $\mathcal{R}_X$ as $v$ has energy $E(t_v) \leq E(\hat{t}_{\mathcal{R}_X})$ (since the fusion can always choose the same labeling as in $\hat{t}_{\mathcal{R}_Y}$). Then, since $t_{\mathcal{R}_X} = \arg\min_{u \in \{u_X : \text{head}(u_X) = X\}} E(t_u)$, where $\{u_X\}$ are the productions of $X$, we have that $E(t_{\mathcal{R}_X}) \leq E(t_v)$ and thus $E(t_{\mathcal{R}_X}) \leq E(\hat{t}_{\mathcal{R}_X})$ and the claim follows. $\square$

# 4 Additional experimental details and results

## 4.1 Additional figures

Figures S1, S2, and S3 show the full matrix of the performance of SFG-PARSE, $\alpha$-expansion, and BP for each measure (minimum energy found, parsing time taken, and mean average pixel accuracy) of the three scenarios (varying the strength of boundary terms, increasing the grammar height, and increasing the number of productions for each nonterminal) described in the main paper.

Figure S1: The (a) best energy, (b) total running time, and (c) resulting semantic segmentation accuracy (mean average pixel accuracy) for belief propagation, $\alpha$-expansion, and SFG-PARSE when varying boundary strength. Each data point is the average value over (the same) 10 images.

Figure S2: The (a) best energy, (b) total running time, and (c) resulting semantic segmentation accuracy (mean average pixel accuracy) for belief propagation, $\alpha$-expansion, and SFG-PARSE when varying grammar height. Each data point is the average value over (the same) 10 images. Missing data points for $\alpha$-expansion indicate that it ran out of memory. Missing data points for BP indicate that it returned infinite energy (left). Low accuracies for grammar height 0 are a result of the grammar being insufficiently expressive.

Figure S3: The (a) best energy, (b) total running time, and (c) resulting semantic segmentation accuracy (mean average pixel accuracy) for belief propagation, $\alpha$-expansion, and SFG-PARSE when varying grammar height. Each data point is the average value over (the same) 10 images. Missing data points for BP indicate that it returned infinite energy (left).

## 4.2 MRF segmentation details

As discussed in the main paper, the energy of each segmentation of a region for a given production is defined by an MRF $E(\mathbf{y}^v, \mathcal{R}_v) = \sum_{p \in \mathcal{R}_v} \theta_p^v(y_p^v; \mathbf{w}) + \sum_{(p,q) \in \mathcal{E}_v} \theta_{pq}^v(y_p^v, y_q^v; \mathbf{w})$. The unary and pairwise terms in $E$ can be defined arbitrarily, as long as the resulting energy is submodular. In our experiments, we define the unary terms for terminals $T \in \Sigma$ as a linear function of the image

features $\theta_p^v(y_p^v = T; \mathbf{w}) = \mathbf{w}_T^\top \phi_p^U$, where $\phi_p^U$ is a feature vector representing the local appearance of pixel $p$. Unary terms for nonterminals $X \in N$ can be defined as $\theta_p^v(y_p^v = X; \mathbf{w}) = w_{pX}^v$, where $w_{pX}^v$ is a (learnable) parameter that specifies how likely this pixel is to be labeled as $X$. This allows each production to encode the regions of the image associated with each of its constituents. Note, however, that in our experiments we do not learn the SFGs and instead simply set the unary terms for nonterminal symbols to 0.

The pairwise terms are also quite flexible, but in our experiments we use the standard contrast-dependent pairwise boundary potential (e.g., Shotton et al. [9]) defined for each production $v$ and each pair of adjacent pixels $p, q$ as $\theta_{pq}^v(y_p^v, y_q^v; \mathbf{w}) = w_v^{\text{BF}} \exp(-\beta^{-1}||\phi_p^B - \phi_q^B||^2) \cdot [y_p^v \neq y_q^v]$, where $\beta$ is half the average image contrast between all adjacent pixels in an image, $w_v^{\text{BF}}$ is the boundary factor that controls the relative cost of this term for each production, $\phi_p^B$ is the pairwise per-pixel feature vector, and $[\cdot]$ is the indicator function, which has value 1 when its argument is true and is 0 otherwise.

### 4.3 $\alpha$-expansion and 3-D MRF details

We compared SFG-PARSE to running $\alpha$-expansion on a flat pairwise MRF and to max-product belief propagation over a multi-level (3-D) pairwise grid MRF. Each label of the flat MRF corresponds to a possible path in the grammar from the start symbol to a production to one of its constituent symbols, etc, until reaching a terminal. In general, the number of such paths is exponential in the height of the grammar. The unary terms are the sum of unary terms along the path and the pairwise term for a pair of labels is the pairwise term of the first production at which their constituents differ. For any two labels with paths that choose a different production of the same symbol (and have the same path from the start symbol) we assign infinite cost to enforce the restriction that an object can only have a single production of it into constituents. Note that after convergence $\alpha$-expansion is guaranteed to be within a constant factor of the global minimum energy [10] and thus serves as a good surrogate for the true global minimum, which is intractable to compute.

The multi-layer MRF for max-product belief propagation (BP) is constructed similarly. The number of levels in the MRF is equal to the height of the DAG corresponding to the grammar used. The labels at a particular level of the MRF are the (production, constituent) pairs that can occur at this height in the (non-recursive) grammar. The pairwise term between the same pixel in two levels is 0 when the parent label's constituent equals the child label's production head, and $\infty$ otherwise. Pairwise terms within a layer are defined as in the flat MRF with infinite cost for incompatible labels (i.e., two neighboring pixels parsed as different productions of the same symbol), unless two copies of that nonterminal could be produced at that level by the grammar.

### 4.4 Details on inference evaluation experiments

We compared the three inference algorithms by varying three different parameters: boundary strength (strength of pairwise terms), grammar height, and number of productions per nonterminal. Each grammar contained a start symbol, multiple layers of nonterminals, and a final layer of nonterminals in one-to-one correspondence with the eight terminal symbols, each of which had a single production that produces a region of pixels. The start symbol had one production for each pair of symbols in the layer below it, and the last nonterminal layer (ignoring the nonterminals for the labels) had productions for each pair of labels, distributed uniformly over this last nonterminal layer.

To generate a grammar, we first defined these parameters (specified below for each experiment), created the symbols in each layer according to these parameters, and then randomly sampled the connectivity of the productions between layers. For example, for each symbol in layer 2 we created the specified number of productions and then for each production we randomly chose two symbols in layer 3 to set as the constituents of this production. Doing this for each symbol in each layer gave us a connected grammar in which each symbol can have multiple parents. For example, in the Grammar Height experiment, each layer had 4 nonterminal symbols in it and each of these symbols had 3 binary productions to the next layer. Thus, each symbol in each middle layer appeared on average as a child in a production (4 symbols * 3 productions * 2 constituents) / 4 symbols = 6 times.

All experiments were run on the same DGX-1 computer running a dual 20-core 2.2 GHz Intel Xeon E5-2698 and 512 GB of RAM. Each algorithm was limited to a single thread.

**Boundary strength.** Increasing the boundary strength of an MRF makes inference more challenging, as individual pixel labels cannot be easily flipped without large side effects. To test this, we constructed a grammar as above with 2 layers of nonterminals (not including the start symbol), each containing 3 nonterminal symbols with 4 binary productions to the next layer. We used a single weight $w_{BF}$ to parameterize all pairwise (boundary) terms in the MRF of every production. Figure S1 plots the mean average pixel accuracy of the parses returned by each algorithm vs. $w_{BF}$ (the x-axis is log-scale). SFG-PARSE returns parses with almost identical accuracy (and energy) to $\alpha$-expansion. BP also returns comparable accuracies, but almost always returns invalid parses with infinite energy (if it converges at all) that contain multiple productions of the same object or a production of a symbol Y even though the pixel is labeled as symbol X.

**Grammar height.** In general, the number of paths in the grammar is exponential in its height, so the height of the grammar controls the complexity of inference and thus the difficulty of parsing images. For this experiment, we set $w_{BF}$ to 20 and constructed a grammar with four nonterminals per layer, each with three binary productions to the next layer. Figure S2 shows the effect of grammar height on total inference time (to convergence or a maximum number of iterations, whichever first occurred). As expected from Proposition 1, the time taken for SFG-PARSE scales linearly with the height of the grammar, which is within a constant factor of the size of the grammar when all other parameters are fixed. Similarly, inference time for both $\alpha$-expansion and BP scaled exponentially with the height of the grammar because the number of labels for both increases combinatorially. Again, the energies and corresponding accuracies achieved by SFG-PARSE were nearly identical to those of $\alpha$-expansion (see Figure 2).

**Productions per nonterminal.** The number of paths in the grammar is also directly affected by the number of productions per symbol. For this experiment, we set $w_{BF}$ to 20 and constructed a grammar with 2 layers of nonterminals, each with 4 nonterminal symbols. Figure S3 shows the effect of increasing the number of productions per nonterminal, which again demonstrates that SFG-PARSE is far more efficient than either $\alpha$-expansion or BP as the complexity of the grammar increases, while still finding comparable solutions (see Figure 3).

### 4.5  Details on model evaluation experiments and grammar induction

To induce a grammar for a particular image, we first constructed 4 segmentations of the image at increasing levels of granularity using the method of Isola et al. [11] and then intersected these regions with the regions from the true labels. We then did the same for 4 other images chosen uniformly at random. The segments from these 5 images define the symbols of the induced grammar and the regions of the segments determine the regions of the symbols in the per-pixel MRF weights. We next created productions between overlapping segments at each pair of neighboring levels of segmentation granularity within the same image. We then generated 3 productions for each symbol by randomly selecting 4 regions in the next level of segmentation granularity that overlapped with the head symbol's region, across all images. Finally, we set the produced terminals of the finest-granularity regions to be those true labels that were expressed anywhere in their region, as these form the bottom of the grammar. For the MRF weights, we set $w_{BF}$ to 5 for all productions and all edges, but used the contrast-dependent pairwise boundary potential defined above to control the strength of the pairwise terms. The deep convolutional neural network features were taken from the layer preceding the softmax in the DeepLab architecture. These were used as the per-pixel unary costs for productions into the corresponding terminal symbol for both the MRF and the SFG experiments. Production weights $\{w_v\}$ were all set to 0, so the SFG energy is entirely determined by the per-pixel unary costs, the pairwise costs, and the structure of the grammar.