[Reviews · NeurIPS 2018]

Reviewer 1



This submission introduces Submodular Field Grammars (SFGs), including its representation and inference algorithms, which is mainly an extension of those previous works, such as [6, 7]. - The novelty/contribution and the practical value of this submission are less than NIPS criteria. - The two conditions stated in line 137~146 significantly limit the extent of applications of the method. - The quantitative evaluation only considers DeepLab and DeepLab+MRF as competitors, which is not convincing. - In line 70~71, it is stated that SFGs are a general and flexible model class that is applicable anywhere grammars or MRFs are used, including social network modeling and probabilistic knowledge bases. Obviously this statement is wrong. How this SFGs can be used for the cases where more general grammars and/or MRFs are involved? - It is stated that a key benefit of the grammar-based formulation is that it enables sub-parse reuse, which enables exponential reductions in inference complexity and better sample complexity. It needs to be validated via experiments to be convincing. - There are various issues in the statements/formulations. For instance, the statement in line 122~124 is inappropriate and y = (y0, . . . , yn) in line 79 is wrong.


Reviewer 2



I thank the authors for their response. I felt the authors gave an good response to my questions. I encourage the authors to more carefully clarify the task since it is a non-standard task if the paper is accepted. ----------- Original review ---------- Summary: Similar to text parsing based on formal grammars, this paper proposes a way to parse images based on formal visual grammars that have a part, subpart structure such as Scene -> Sky, Ground. The key problem is that splitting the image into *arbitrarily-shaped* pixel regions to associate with the production rules is computationally difficult in general. This paper proposes to associate formal grammar production rules with submodular Markov random fields (MRF). The submodular structure of the associated MRF allows for fast inference for a single rule into arbitrarily-shaped subregions and a dynamic-programming-like algorithm for parsing the entire image structure. The experimental results show that the method is indeed much faster than previous methods. Pros: 1) Well-written and easy to read even though some of the details are fairly technical. 2) Excellent speed results with competitive or better accuracy. 3) Novel combination of MRFs and formal grammars for images Cons: 1) The specific task (i.e. input data/parameters and output) seemed a little confusing (note that I am not particularly familiar with the parsing task especially for images). Is the task a hierarchical version of semantic segmentation? Are the raw image pixels given to the algorithm or just semantic features like segmentation label? In one place the energy seems to depend on the raw pixels (Sec. 2.1) but later the energy is denoted to only depend on y (Sec. 4.1); maybe this is just a notation thing but I wanted to check. 2) The experiments are also a little confusing but maybe this is just because the task is not a classic task. How are the DeepLab unary features used? Are they used instead of the raw pixel values? 3) What would be an example of a non-tree-structured grammar (see lines 66-68)? It would be good to more clearly describe the difference from [6,7 ref in paper] that are similar but for tree-structured grammars. Minor comments: The illustrations were very helpful in making the ideas concrete. I would suggest presenting the theorem/propositions first and then giving comments on them instead of defining them after giving comments.

Reviewer 3



Image parsing is an important topic in computer vision. However, unlike grammar parsing, image parsing is hard due to exponential many ways of splitting an image region. The authors attacking this problem by associate each splitting with an submodular Markov Random Field (MRF), and an approximate algorithm of MAP parsing is also developed with such structure. Pros: (1) Allows arbitary shape of regions instead of the very restricted setting (e.g. regions must bu rectangular). Allowing arbitary shape of regions significantly makes the algorithm more practical, as objects can have arbitary shape rather than rectangular. (2) The hierarchical structure of the proposed method results in an efficient parsing algorithm. (3) Convergence of the parsing algorithm is guaranteed, and the time complexity of the algorithm is reasonable. Cons: (1) Maybe the experiment part. The current experiment is performed on a small dataset. If the authors can get more results on larger dataset, if would be much better.